# Cross-Reactive Carbohydrate Determinant in *Apis mellifera*, *Solenopsis invicta* and *Polybia paulista* Venoms: Identification of Allergic Sensitization and Cross-Reactivity

**DOI:** 10.3390/toxins12100649

**Published:** 2020-10-08

**Authors:** Débora Moitinho Abram, Luís Gustavo Romani Fernandes, Amilcar Perez-Riverol, Márcia Regina Brochetto-Braga, Ricardo de Lima Zollner

**Affiliations:** 1Laboratory of Translational Immunology, Medicine School, FCM, University of Campinas (UNICAMP), Rua Tessália Vieira de Camargo, nº 126, 13083-888, Cidade Universitária “Zeferino Vaz”, Campinas SP 13500, Brazil; deboramoitinhoabra@gmail.com (D.M.A.); lgrf@fcm.unicamp.br (L.G.R.F.); 2Center for the Study of Social Insects, Department of General and Applied Biology, Institute of Biosciences of Rio Claro, Institute of Biosciences of Rio Claro, São Paulo State University, Rio Claro SP 13500, Brazil; aperezriverol@gmail.com (A.P.-R.); mrb.braga@unesp.br (M.R.B.-B.)

**Keywords:** Hymenoptera venom allergy, cross-reactive carbohydrate, *Polybia paulista*, bromelain

## Abstract

Allergic reactions to Hymenoptera venom, which could lead to systemic and even fatal symptoms, is characterized by hypersensitivity reactions mediated by specific IgE (sIgE) driven to venom allergens. Patients multisensitized to sIgE usually recognize more than one allergen in different Hymenoptera species. However, the presence of sIgE directed against Cross-Reactive Carbohydrate Determinant (CCD), which occurs in some allergens from Hymenoptera venom, hampers the identification of the culprit insects. CCD is also present in plants, pollen, fruits, but not in mammals. Bromelain (Brl) extracted from pineapples is a glycoprotein commonly used for reference to sIgE-CCD detection and analysis. In sera of fifty-one Hymenoptera allergic patients with specific IgE ≥ 1.0 KU/L, we assessed by immunoblotting the reactivity of sIgE to the major allergens of *Apis mellifera*, *Polybia paulista* and *Solenopsis invicta* venoms. We also distinguished, using sera adsorption procedures, the cases of CCD cross-reaction using Brl as a marker and inhibitor of CCD epitopes. The presence of reactivity for bromelain (24–28 kDa) was obtained in 43% of the patients, in which 64% presented reactivity for more than one Hymenoptera venom in radioallergosorbent (RAST) tests, and 90% showed reactivity in immunoblot analysis to the major allergens of *Apis mellifera*, *Polybia paulista* and *Solenopsis invicta* venoms. Sera adsorption procedures with Brl lead to a significant reduction in patients’ sera reactivity to the Hymenoptera allergens. Immunoblotting assay using pre- and post-Brl adsorption sera from wasp-allergic patients blotted with non-glycosylated recombinant antigens (rPoly p1, rPoly p5) from *Polybia paulista* wasp venom showed no change in reactivity pattern of sIgE that recognize allergen peptide epitopes. Our results, using Brl as a marker and CCD inhibitor to test sIgE reactivity, suggest that it could complement diagnostic methods and help to differentiate specific reactivity to allergens’ peptide epitopes from cross-reactivity caused by CCD, which is extremely useful in clinical practice.

## 1. Introduction

Allergic reaction to insect venom is one of the most common causes of anaphylaxis [1] and may occur in up to 42% of accident cases [2]. Among them, the Hymenoptera superfamilies represented by Apoidea (bees), Vespoidea (wasps) and Formicidae (ants) are clinically relevant [3,4,5]. Accidents with the stings of these insects cause symptoms such as pain, itching and swelling in all cases, and, in some allergic patients, specific reactions are observed as exuberant local reactions, systemic reactions and fatal reactions [6].

Allergenic proteins present in Hymenoptera venoms may trigger a type I hypersensitivity reaction, mediated by antigen-specific immunoglobulin E (IgE), which is a risk factor for subsequent systemic reactions (SR) [7]. However, the level of venom-specific IgE does not correlate with the severity of the SR, and some patients with poorly detectable specific venom-IgE can have near-fatal anaphylaxis. Furthermore, sera from sensitized patients may present multiple reactivities (30–50%) and cross-react with another Hymenoptera venom (60–70%) [7,8]. These reactions may occur due to several factors such as elevated specific IgE levels, independent of the sensitization to each venom; IgE antibodies with cross-reactivity against homologous proteins present in venoms or by the binding of IgE antibodies to cross-reactive carbohydrate determinants (CCD) [9,10,11].

CCDs are present in glycosylated proteins, and the glycosylation process occurs in the endoplasmic reticulum and in the Golgi complex by the addition of carbohydrates to specific amino acid residues of proteins [12]. In hymenopteran proteins, N-glycosylation occurs, in which oligosaccharides (glycans) bind to the nitrogen atom present in the amino acid residues of asparagine (Asn). In plants and hymenopterans, the N-glycans added are similar; for example, α1,3-fucose and, less frequently, α1,6-fucose (main structure known as MMF^3^F^6^), being epitopes highly immunogenic and capable of inducing the production of IgE antibodies, [9,12,13,14].

Although CCD is considered an immunogenic molecule, it is clinically irrelevant, because there are no reports of allergic reactions caused as a result of immune response triggered to these epitopes [15]. This lack of allergic reactions can be attributed to the fact that the CCD is a monovalent structure, which does not allow cross-link between the CCD and the anti-CCD IgE linked to FcεRI receptors on the mast cell, thus not causing mast cell degranulation and release of inflammatory mediators such as vasoactive amines and cytokines [16]. However, the anti-CCD IgE is relevant to the hymenopteran allergy diagnosis, since more than 20% of allergic patients develop these antibodies, which could be related to the generation of false-positive results on RAST tests [6].

Advances in the analysis of hymenopteran venoms proteins showed that some allergens, such as the Hyaluronidase [17] and phospholipase A2 from *Apis mellifera* [18] and phospholipase A1 from *Solenopsis invicta* [19], present CCDs in their structure. It is also known that *Polybia paulista* allergenic proteins, phospholipase A1 and antigen 5 are glycosylated proteins, so they do not have CCD epitopes, such as antigen 5 from *Solenopsis invicta* [20,21].

CCDs are molecular structures also found in bromelain (Brl), an enzyme of approximately 24 kDa obtained from the crude and aqueous extract of the pineapple (*Ananas comosus)*, which has two N-glycans, β1,2-xylose and α1,3- fucose, with a structure called MUXF^3^. Due to the similarity between this structure and the CCD epitopes present in hymenopteran allergens, Brl can be a useful tool to differentiate patients that present only anti-CCD IgE from those that present specific recognition of protein epitopes [9,22,23,24,25]. Therefore, this study aims to verify cases of CCD cross-reaction using bromelain as a marker and inhibitor of CCD epitopes in patient sera presenting IgE isotype-specific antibodies (≥1.0 kU/L) to *Apis mellifera*, *Polybia paulista* and *Solenopsis invicta* venom.

## 2. Results

### 2.1. Electrophoretic Profile of the Allergens Obtained in the Hymenoptera Venoms Extracts and Bromelain

The electrophoresis of the venoms extracts showed bands corresponding to the following proteins: Hyaluronidase (Hyal): 45 kDa; Phospholipase A2 (PLA2): 16 kDa and trypsin inhibitor rich in cysteine (Api m6): 8 kDa from *Apis mellifera*; Phospholipase A1 (PLA1): 34 kDa and Antigen 5 (Ag 5): 23 kDa from *Polybia paulista*; and Phospholipase A1 (PLA1B): 35 kDa and Antigen 5 (Ag 5): 24 kDa from *Solenopsis invicta*. Electrophoresis showed the Brl as a band around 24–28 kDa. Additionally, we performed electrophoresis using the recombinant forms of PLA1 and Ag 5 since they were used in some blotting experiments described in item 2.5 (Figure 1).

### 2.2. Patient’s Specific IgE Reactivity Profile

Table 1 shows the data analysis of specific IgE (sIgE) to Hymenoptera venoms (bee, wasp and ant) obtained from RAST tests of fifty-one allergic patients. Among the total number of patients, eighteen (35%) showed sIgE ≥ 1.0 KU/L for the three venoms studied, indicating a multiple reactivity profile, ten (20%) of them showed double reactivity, and twenty-three (45%) reactivity for only one of the venoms (eight for bees, six for wasp, and nine for ants).

Immunoblotting analysis showed that, among all fifty-one patients, only one patient (pt 22) did not show serum reactivity to allergens presented in the *Apis mellifera* venom extract. The other fifty showed reactivity to PLA2 protein. Additionally, twelve of them also reacted to Hyal., and thirty-eight reacted to the Api m6 protein. Only ten patients (pt 1; pt 3; pt 5; pt 6; pt 7; pt 12; pt 14; pt 16; pt 24; pt 25) presented reactivity to all three allergens (Table 2 and Table 3, Figure A1 and Figure A2).

On the other hand, when we analyzed the serum reactivity to *Polybia paulista* venom extract, we observed that just three patients (pt 29, pt 38 and pt 46) did not have serum reactivity to either of the two studied allergens (PLA1 and Ag 5). Furthermore, thirty-four patients showed serum reactivity to both allergens, ten patients presented serum reactivity to PLA1 (pt 11; pt 13; pt 18; pt 19; pt 23; pt 28; pt 30; pt 34; pt 43; pt 47), and four patients had serum reactivity only for Ag5 (pt 33; pt 44; pt 45; pt 48) (Table 2 and Table 3, Figure A1 and Figure A2).

Finally, in the analysis of serum reactivity profile to *Solenopsis invicta* venom extract, we observed that sera from only three patients (pt 4; pt 23; pt 38) did not react to any of the studied proteins (PLA1 and Ag5), forty-one patients had serum reactivity to both allergens, six patients had serum reactivity to PLA1B only (pt 1; pt 2; pt 3; pt 11; pt 25; pt 32), and one patient had serum reactivity only to Ag5 (pt 9) (Table 2 and Table 3, Figure A1 and Figure A2).

### 2.3. CCDs sIgE Reactivity Profile

Twenty-two sera from the studied population recognized Brl in the immunoblot assays in opposite to twenty-nine sera that did not react. Among the patients’ serum that recognize Brl, 64% (14/22) showed reactivity for more than one Hymenoptera venom, and 36% (8/22) presented reactivity for just one, as demonstrated by the presence of sIgE in the RAST tests. On the other hand, in patients’ sera without recognition of Brl, 48% (14/29) showed reactivity for more than one Hymenoptera venom, and 51.7% (15/29) presented reactivity for just one (Table 4). The results indicate that in the group that reacts to Brl, the presence of sIgE is predominant for two or even for the three studied venoms. In contrast, in the non-reactive-to-bromelain group, there is a predominance of sera with sIgE for just one venom.

However, the analysis of immunoblot assays showed that 90% (20/22) of patients with anti-CCD antibodies in the sera showed reactivity for all Hymenoptera venoms tested (Table 2 and Figure A1), whereas 86% (25/29) of patients without anti-CCD antibodies had reactivity for all three Hymenoptera venoms (Table 3 and Figure A2). These results could indicate a similar sIgE reactivity profile when comparing patients with anti-CCD antibodies with those who did not present them.

### 2.4. Bromelain Adsorption Effect on Patient’s Sera

To eliminate non-specific sIgE reactivity, due to the presence of anti-CCD antibodies presented in patient’s serum, we used an adsorption protocol incubating sera with Brl. Immunoblotting procedures were then conducted with serum previously adsorbed and blotted with *Apis mellifera*, *Polybia paulista* and *Solenopsis invicta* venom extracts. The intensities of the pre- and post-adsorption bands are presented in Table 2, and the images of the immunoblot assays are shown in Figure A1.

We observed a significant reduction in band intensity in all patients’ sera tested and blotted to all Hymenoptera species venoms used in this study (pre-adsorption versus post-adsorption, Table 2). In immunoblot assays with *Apis mellifera* venom, post-adsorption sera from pt 1, pt 4, pt 5, pt 9, pt 15, pt 16, pt 17 and pt 19 did not react to any of the allergens studied and pt 22 remains negative. The other patients maintain reactivity to the allergens, despite the change in the intensity of the bands. Similarly, in immunoblot assays with *Polybia paulista* and *Solenopsis invicta* venom, there is also a reduction in the intensity of the bands in the post-adsorption sera and the pt 1, pt 4, pt 5, pt 6, pt 8, pt 11, pt 12, pt 13, pt 14, pt 16 and pt 19 sera stopped reacting to the venom allergens from *Polybia paulista* and the sera from pt 4, pt 5, pt 6, pt 8, pt 17, pt 20 and pt 21 no longer react to the allergens of *Solenopsis invicta* venom. Interestingly, the sera from pt 4 and pt 5 stopped reacting to all venoms studied after adsorption with bromelain, although both patients have sIgE for the three venoms, as shown in RAST results (Table 1).

We also used immunoblotting assays to investigate the effect of adsorption with Brl in the serum from patients who did not present anti-CCD antibodies (Figure A2). As shown in Table 3, we found evidence of a reduction in the band’s intensities, however, with a lower frequency. In immunoblots with *Apis mellifera* venom, serum from pt 26, pt 32, pt 38 and pt 43 stopped reacting to all allergens, whereas, for *Polybia paulista* venom, sera from pt 29, pt 38 and pt 46 remain negative, and sera from pt 24, pt 25, pt 27, pt 28, pt 31, pt 34, pt 35, pt 37, pt 39, pt 40, pt 41, pt 42, pt 43, pt 46, pt 47, pt 49 and pt 51 stopped reacting to the two allergens presented in this venom. On the other hand, the immunoblotting analysis with *Solenopsis invicta* venom showed that sera from pt 26, pt 32, pt 41, pt 42, pt 43, pt 44, pt 46 and pt 47 stopped reacting to the allergens in the venom extract and sera from pt 23 and pt 38 remain non-reactive to these allergens.

### 2.5. IgE-Specific Reactivity to Non-Glycosylated Recombinant Antigens (rPoly p1, rPoly p5) from Polybia Paulista Wasp Venom

To verify an eventually allosteric binding effect in sIgE that recognize peptides epitopes, during the adsorption of patient’s sera with Brl, sera of the six patients with IgE reactive to *Polybia paulista* venom and Brl were selected. Immunoblotting was performed with the *Polybia paulista* venom extract and with the recombinant allergens rPoly p 1 (PLA1) and rPoly p 5 (Ag 5). As shown in Figure 2, there is no change in the sera reactivity pattern to the recombinant allergens when comparing pre- and post-adsorption sera.

## 3. Discussion

The presence of cross-reactive carbohydrates (CCD) in the allergens of Hymenoptera venoms is considered essential to serum antibody cross-reactivity that could be misinterpreted as multiple sensitizations hampering the correct diagnosis of the culprit insect in allergic patients [9,10,11]. This phenomenon could be observed in up to 59% of bee and wasp venoms allergic patients [19,26]. It is pondered that IgE directed to CCD is the major cause of multiple sensitization detection [27,28]. Here, we demonstrated changes in the sIgE reactivity profile comparing immunoblot results with sera from patients allergic to *Apis mellifera*, *Polybia paulista* and *Solenopsis invicta* venoms pre- and post-adsorption with Brl, a plant glycoprotein known as a source of N-glycans that mimics the CCDs presented in some allergens’ composition of the Hymenoptera venoms.

Firstly, we demonstrated that the Hymenoptera venom extracts, used as the antigenic source in the immunoblot procedures, present the major allergens typically found in their respective species [20]. Seven proteins were found with relevant reactivity to the patient’s sera: Hyaluronidase (45 kDa), PLA2 (16 kDa), Api m6 (8 kDa) in *Apis mellifera* venom extract, PLA1 (34 kDa) and Ag 5 (23 kDa) in *Polybia paulista* venom, and PLA1B (35 kDa) and Ag 5 (24 kDa) in *Solenopsis invicta* venom (Figure 1). As described in previous work, the recombinant forms of PLA1 and Ag 5 from *Polybia paulista* used in some experiments presented similar molecular weight and immunological reactivity when compared with the native allergens [29,30].

The analysis of sIgE reactivity profile of the fifty-one patients’ serum showed the prevalence of multiple sensitizations, characterized by the presence of sIgE for more than one Hymenoptera species venom. Immunoblot assays using chemiluminescent detection showed that forty-nine of the fifty-one patients (96%) have sIgE able to recognize allergens of more than one species. In contrast, RAST data showed that only twenty-eight patients (55%) showed this same reactivity pattern. The difference in the sIgE reactivity found when we are comparing both sIgE detection methods could be explained by the sensitivity of each method. Consistently with this, Mosbech et al. [31] demonstrated that multi-allergosorbent chemiluminescent-based detection technique presented higher sensitivity in comparison with RAST test to detect sIgE to ten different types of inhalant allergens. Furthermore, immunoblot procedures can detect and identify other allergens in venom extracts, being, therefore, more sensitive, and adequate to the objectives of the present work.

Bromelain has been used in several studies to elucidate the presence of CCD in Hymenoptera allergens and to identify anti-CCD antibodies in sera from allergic patients [18,32,33,34]. Among these studies, Tretter V et al. [18] used Brl to identify the presence of N-glycans in one hundred twenty-two patients with bee-specific IgE; these authors observed reactivity in thirty-four sera of the allergic patient. This study also concluded that the interaction between IgE and PLA2 N-glycan could be inhibited by bromelain due to the presence of α1,3-fucose, also present in insect venoms. Eržen et al. [32] confirmed CCD cross-reactivity by quantifying sIgE for MUFX^3^ (glycoprotein structure extracted from Brl), being present in 43% of cases of double-sensitization to bee and wasp venom. Jappe et al. [33] found that the sera of patients with bee-specific and yellow-jacket-specific IgE, when tested for Brl, also showed reactivity to this protein in 67% of the cases. Here, we used bromelain to trace the presence of anti-CCD IgE (pre-adsorption assays) in Hymenoptera-allergic patients’ serum. We showed a band reactive to bromelain in twenty-two from the fifty-one tested sera (43%), which is consistent with the data in the literature [32].

Using a similar strategy, Rodríguez-Pérez et al. [35] analyzed cross-reactivity among the nematode *Anisakis spp.* allergens and those from wasp venom. Using bromelain to inhibit anti-CCD IgE, through the previous incubation of the sera with this source of CCDs, it was demonstrated in the immunoblot assay the loss of the patient’s serum reactivity in cases where CCD was previously detected in the chemical structure of the allergens. Hemmer et al. [34], using CCD inhibitors (including MUXF^3^) to investigate the cases of double-sensitization, showed in immunoblot assays a significant change in sera reactivity after inhibition procedures, demonstrating that previous serum adsorption procedures with CCD inhibitors could be a promising strategy for the tracing of CCD cross-reactivity. Consistently with this, in our study, we observed a significant reduction in the sera reactivity to all allergens studied, after the sera adsorption with Brl.

The change in sera reactivity pattern after Brl adsorption procedures, observed in our study, demonstrated that there was a partial or total reduction not only in the 22 patients reactive to Brl but also in the group of patients who did not react previously to Brl. Interestingly, some serum stopped reacting to PLA1 and Ag 5 from *Polybia paulista* or to Ag5 from *Solenopsis invicta* after Brl adsorption procedures. Perez-Riverol et al. [36] in recent studies have shown that PLA1 and Ag 5 allergens from *Polybia paulista* are glycosylated proteins and therefore do not have CCD, as in the Ag 5 from *Solenopsis invicta* [20]. Thus, this reduction in sera reactivity could not be related to α1,3 – fucose epitopes recognition. Our hypothesis to explain the findings is the possible association with the allosteric physicochemical effect generated during the adsorption with bromelain. The allosteric effect occurs when there are molecules capable of causing conformational changes in the structure of a protein, thus being able to induce modifications in the antigen-binding sites of the antibodies, for example.

We observed that the loss of sera reactivity, after adsorption with Brl, occurs in those that previously had low reactivity (+) to the allergen, so after the adsorption procedure, this reactivity tends to disappear, probably due to an allosteric effect, as mentioned above. This effect can also explain the fact that in some patients’ sera (pt 20, pt 21, pt 26, pt 28, pt 33, pt 44 and pt 48) there was an increase in sera reactivity, or they even started to react to allergens that had not been previously detected in the immunoblot assay (pt 20 and pt 21 with Api m 6 from *Apis mellifera* and pt 33, pt 44 and pt 48 with PLA1 from *Polybia paulista*), after the adsorption procedure with Brl.

However, in cases where there is a previous high reactivity (+++) to the allergens, the signal decreases after adsorption with Brl, which should probably be associated with the inhibition of anti-CCD antibodies, as described by Hemmer et al. [34] and in other by allosteric interference. Similarly, in the case of the glycosylated allergens Hyaluronidase, PLA2, Api m 6 and PLA1B, the reduction in sIgE reactivity after adsorption may be related to the presence of CCD epitopes. Additionally, a suitable explanation should be a previous sensitization to CCD developed with contact with other allergens that also carry these epitopes, such as pollen allergen [37], which has a CCD structure very similar to those found in the Hymenoptera allergens. Conversely, for some patients’ sera (per ex., p35 and p51), the signal reduction after Brl adsorption could not be explained by anti-CCD because Poly p 1 and Poly p 5, do not have CCD in their composition, so in this case, the allosteric interference represents the most likely explanation.

Additionally, we observed no change in the serum reactivity pattern to the recombinant allergens when comparing pre- and post-adsorption sera. Since the recombinant allergens are non-glycosylated proteins, we demonstrated that the adsorption procedures did not interfere with the sIgE antibodies that recognize peptides epitopes. Finally, despite the reduction in sIgE reactivity intensity, several patients still maintain reactivity to the allergens from the three Hymenoptera venoms, even after adsorption procedures, indicating that these reactivities may be caused by another reason, such as similarity of peptides epitopes or cases of multiple independent sensitivities, and not exclusively by CCD recognition.

## 4. Conclusions

The anti-CCD IgE presence, despite being clinically irrelevant, can interfere with the proper diagnosis, considering the reactivity to the venom, and consequently with the prescription of adequate immunotherapy. The diagnosis of Hymenoptera venom allergy is dependent on several factors, such as a clinical classification of anaphylaxis, skin tests and specific serum IgE quantification. Our results, using bromelain as a marker and CCD inhibitor, alter the IgE reactivity. Thus, it supplements diagnostic methods and helps to differentiate clinical cases with cross-reactivity caused by CCD, being useful and promoting improvement in clinical practice.

## 5. Materials and Methods

### 5.1. Patients

Sera from 51 Hymenoptera venom-sensitized patients were selected considering specific IgE ≥ 1.0 kU/L (RAST from medical archive), regardless of sex or age. They were obtained from the Ambulatório de Anafilaxia—Hospital de Clínicas, Faculdade de Ciências Médicas, Universidade Estadual de Campinas—UNICAMP. The FCM-UNICAMP Ethics Committee approved the study of under n. 80822817.3.0000.5404, Date: 2 January 2018. Informed consent was obtained in written form from all participants of the study, and participation was voluntary. The samples were stored at −30 ^°^C until use.

### 5.2. Venoms Extracts and Bromelain Solution Preparation

The collection of specimens *Polybia paulista* and *Solenopsis invicta* were authorized by SISBIO under number 60025-2 and duly registered in SISGEN under number AEE0F92. Venom glands were extracted from the *Polybia paulista* specimens and pulverized in liquid nitrogen, and then diluted in saline (NaCl)/ bicarbonate (NaHCO_3_) solution (85 mM/33 mM). *Solenopsis invicta* venom was obtained according to the protocol previously described by Fox et al. (2013) [38]. Powdered *Apis mellifera* (apitoxin) venom was obtained commercially from the National Cooperative of Beekeepers of Minas Gerais (CONAP, Nova Lima, MG-Brazil) and 10 g of the extract were diluted in 40 mL of saline (NaCl)/bicarbonate (NaHCO_3_) solution (85 mM/33 mM). The lyophilized bromelain was obtained commercially (Sigma, St. Louis, MO, USA), and extracts were prepared by diluting 10 mg in 1 mL of deionized water. Protein quantifications were performed using the colorimetric methods of Hartree [39] (*Apis mellifera* and *Polybia paulista*) and Bradford [40] (Bromelain and *Solenopsis invicta*).

### 5.3. Recombinant Allergens (rPoly P1 and rPoly p5) Obtention

The recombinant allergens rPoly p 1 and rPoly p 5 were obtained as described by Perez-Riverol et al. (2016) [29] and Bazon et al. (2017) [30], respectively. Briefly, the rPoly p 1 was obtained expressing the PLA1 cDNA cloned in PET-28a vector and expressed in *E. coli* BL21 (DE3) cells using the previously described protocol [29]. The rPoly p 5 were obtained, expressing the Ag5 cDNA cloned in pPICZαA vector and expressed in *P. pastoris* X-33 cells, using the previously described protocol [30]. For the rPoly p 1 and rPoly p 5 purification, the soluble fractions were applied to a prepacked column HisTrap HP™ (Ni^+2^ Sepharose™ High Performance; GE Healthcare, Danderyd, Sweden), according to the manufacturer’s instructions, followed by SDS-PAGE (15%) analysis for monitoring the efficiency of the purification process.

### 5.4. Adsorption Tests with Bromelain and Serum

The sera adsorption was carried out in 96-well ELISA plates (Spectra Plate 96HB, Waltham, MA, USA), previously coated (0.1 µg/well) with bromelain. The plates were blocked with 300 µL/well of the blocking solution (PBS-10% fetal bovine serum, Vitrocell, Campinas-SP, Brazil) and incubated for 1 h at 37 °C in a humid chamber. After this period, diluted sera (1:50 or 1:100) were incubated for 18 h (100 µL/well) at a temperature of 4 °C in a humid chamber, and subsequently that sera were collected for immunoblotting assays.

### 5.5. Electrophoresis and Immunoblot Assay

SDS-polyacrylamide gel electrophoresis (SDS-PAGE) 10–20% was performed according to Laemmli, 1970 [41], using a Mini-Protean^®^ Tetra Cell System (BioRad, Hercules, CA, USA), followed by staining with either Coomassie Brilliant Blue R-250 (Sigma-Aldrich) or silver stain. Briefly, 100 μg of *Apis mellifera* or *Polybia paulista* venom extracts, and 50 μg of *Solenopsis invicta* and Bromelain, were submitted to SDS-PAGE and electrotransferred to a 0.45 µm nitrocellulose membrane (Bio-rad) using a semi-dry system (Trans-Blot^®^ SD Semi-Dry Electrophoretic Transfer Cell, Bio-Rad, CA, USA). After blocking with 20 mM Tris-HCl, 150 mM NaCl, pH 7.4, with 0.5% Tween-20 (Sigma-Aldrich, St. Louis, MO, USA) (TBS-T wash solution) and 3% non-fat dried milk (block solution) for 2 h at room temperature under slow agitation on a Rocker II™ Platform mixer (Boekel Scientific, Feasterville-Trevose, PA, USA), the membranes were washed (3 times with TBS-T wash solution). The membranes were cut into strips of approximately 5 mm and incubated for 18 h with the patient’s sera diluted at 1:50 or 1:100 in TBS-T. Immunodetection was performed using anti-human IgE (ε-chain specific) peroxidase conjugate antibody (Sigma-Aldrich) (1:3000). The bands were visualized in Image Quant 400 (GE Healthcare, Uppsala, Sweden) using the chemiluminescent substrate Luminata™ Forte Western HRP substrate (Millipore, Billerica, MA, USA). The results of the immunoblots were analyzed qualitatively by determining the intensity of the venom and bromelain protein (CCD) bands, established from a pre-established model and represented by +; ++, and +++, as shown in Figure 3.

## Figures and Tables

**Figure 1 toxins-12-00649-f001:**
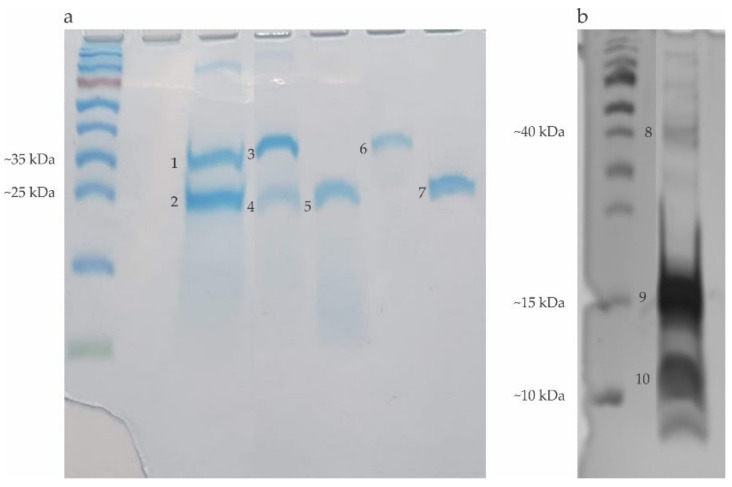
Electrophoresis in gradient gel (10–20%), polyacrylamide with sodium dodecyl sulfate (SDS-PAGE) with crude venoms, recombinants and bromelain (Brl). (**a**) (Coomassie blue staining): 1: PLA1B; 2: Antigen 5 (*Solenopsis invicta*); 3: PLA1; 4: Antigen 5 (*Polybia paulista*); 5: Brl; 6: Recombinant PLA1; 7: Recombinant antigen 5. (**b**) (Silver staining): 8: Hyaluronidase; 9: PLA2; 10: Api m 6 (*Apis mellifera*).

**Figure 2 toxins-12-00649-f002:**
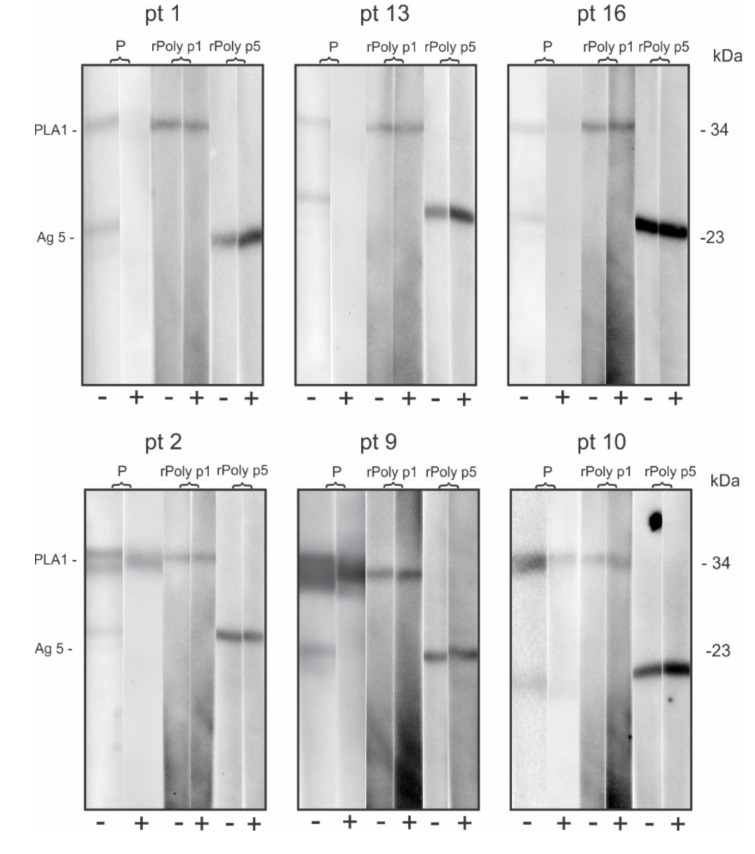
Immunoblot analysis of six patients’ sera (pt) pre (-) and post (+) adsorption with Brl blotted with venom extract and recombinant allergens (rPoly p1: PLA1 34 kDa, and rPoly p 5: Antigen 5 23 kDa) from *Polybia paulista*.

**Figure 3 toxins-12-00649-f003:**
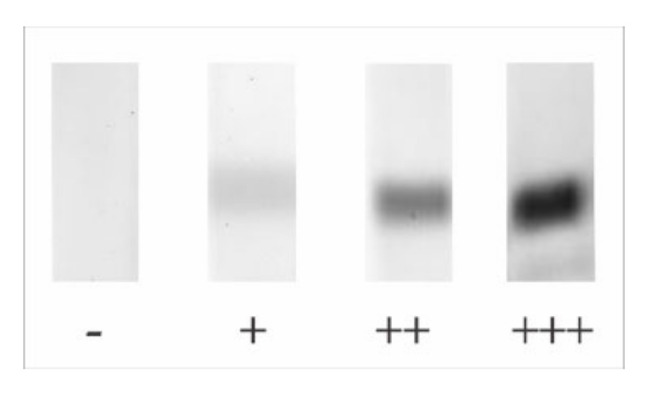
The image is representing the semi-quantitative analysis of immunoblotting assays.

**Table 1 toxins-12-00649-t001:** Total IgE and RAST results from fifty-one Hymenoptera venom allergic patients.

Patient	Age (Years)	Sex	Total IgE	RAST Bee	RAST Wasp	RAST Ant
1	28	m	76	12.5	5.4	1.6
2	3	m	1228	10.8	1.2	15.9
3	9	m	1878	19.6	16.8	11.9
4	8	m	841	1.8	1.0	24.0
5	7	m	433	7.4	2.1	4.1
6	7	f	277	27.0	15.2	4.7
7	32	m	2979	13.9	42.2	61.6
8	7	m	136	2.3	6.0	7.0
9	20	m	462	4.9	2.7	6.4
10	26	m	485	24.0	4.8	7.6
11	24	f	210	10.9	15.0	neg
12	8	f	n.d.	3.3	neg	2.2
13	52	f	n.d.	24.7	1.5	neg
14	33	m	n.d.	24.8	3.5	neg
15	40	f	500	neg	12.3	neg
16	10	m	113	4.7	neg	neg
17	7	f	n.d.	neg	neg	7.7
18	8	m	1023	neg	neg	8,9
19	7	m	264	neg	1.0	neg
20	6	f	13	neg	neg	2.7
21	7	f	n.d.	neg	1.3	neg
22	38	m	35	1.1	neg	neg
23	55	m	n.d.	18.9	11.2	24.9
24	55	m	n.d.	6.1	4.7	1.0
25	4	m	1589	>100	4.7	31.0
26	42	f	183	3.8	1.0	5.9
27	8	m	n.d.	19,3	2,4	11.3
28	8	m	1459	5.7	2.1	7.9
29	17	m	764	2.4	11.5	1.0
30	17	m	207	5.0	10.7	10.8
31	63	f	n.d.	1.0	3.2	neg
32	52	m	62	5.3	6.1	neg
33	4	m	116	9.1	neg	3.2
34	9	f	n.d.	9.7	neg	>100
35	46	m	n.d.	2.9	4.0	neg
36	12	m	653	3.8	neg	1.4
37	52	m	n.d.	2.5	neg	neg
38	51	m	n.d.	1.0	neg	neg
39	30	m	n.d.	9.2	neg	neg
40	45	m	149	68.4	neg	neg
41	44	m	n.d.	1.7	neg	neg
42	22	f	106	neg	neg	3.1
43	10	m	372	neg	neg	24.5
44	7	m	n.d.	neg	neg	36.1
45	20	f	344	neg	neg	44.9
46	5	m	424	neg	neg	26.9
47	34	m	217	neg	1.1	neg
48	69	f	40	neg	1.1	neg
49	57	m	n.d.	neg	1.6	neg
50	19	f	n.d.	12.8	neg	neg
51	9	m	487	neg	neg	23.1

1. sIgE = kU/L; 2. n.d. = not determined.

**Table 2 toxins-12-00649-t002:** The relative intensity of immunoblotting assay: patient’s serum with anti-CCD.

		*Apis mellifera*			*Polybia paulista*		*Solenopsis invicta*	
	Patient	Hyal.	PLA2	Api m6	PLA1	Ag 5	PLA1	Ag 5
Pre-adsorption	1	++	+++	+++	++	++	++	-
Post-adsorption		-	-	-	-	-	+	-
	2	-	++	++	++	+	+	-
		-	+	+	++	-	+	-
	3	+	++	++	+++	+	+	-
		-	+	+	+++	-	+	-
	4	-	+	-	+	+	-	-
		-	-	-	-	-	-	-
	5	+	++	++	++	++	+++	++
		-	-	-	-	-	-	-
	6	++	++	+	+	+	+++	++
		-	+	-	-	-	-	-
	7	++	++	+	+	++	+++	+++
		-	+	-	+	++	++	+
	8	-	++	++	+	+	++	+
		-	+	-	-	-	-	-
	9	-	++	+++	+++	++	-	+
		-	-	-	+++	-	-	+
	10	-	++	-	+++	+	+++	+++
		-	+	-	++	-	++	++
	11	++	+	-	++	-	++	-
		+	-	-	-	-	++	-
	12	++	++	++	++	++	+++	+
		-	+	+	-	-	++	-
	13	-	+	+	++	-	+	+
		-	+	-	-	-	+	+
	14	+	++	++	+++	+	+	+
		+	++	-	-	-	+	+
	15	-	+	-	++	+	++	++
		-	-	-	+	+	+	+
	16	++	++	++	+	+	+++	+
		-	-	-	-	-	+	-
	17	-	++	-	+	+	++	++
		-	-	-	+	+	-	-
	18	-	+	++	++	-	++	++
		-	+	-	+	-	+	-
	19	-	+	-	++	-	++	+++
		-	-	-	-	-	+	+
	20	-	+	-	++	+	+	++
		-	+	+	++	+	-	-
	21	-	+	-	+	++	++	++
		-	+	+	+	+	-	-
	22	-	-	-	++	+	++	++
		-	-	-	+	-	+	+

+ low reactivity; ++ medium reactivity; +++ high reactivity; - non-reactivity; CCD: cross-reactive carbohydrate determinants.

**Table 3 toxins-12-00649-t003:** The relative intensity of immunoblotting: patient’s serum without anti-CCD.

		*Apis mellifera*			*Polybia paulista*		*Solenopsis invicta*	
	Patient	Hyal.	PLA2	Api m 6	PLA1	Ag 5	PLA1	Ag 5
Pre-adsorption	23	-	++	+	++	-	-	-
Post-adsorption		-	+	-	++	-	-	-
	24	+	++	+	++	++	+	+
		-	+	+	-	-	+	-
	25	+	++	++	+	+	+	-
		-	-	-	-	-	+	-
	26	-	+	+	+++	+	+	+
		-	-	-	+++	-	-	-
	27	+	++	-	+	+	++	++
		-	+	-	-	-	+	-
	28	-	++	++	+	-	+	+++
		-	+	-	-	-	++	+++
	29	-	++	++	-	-	+++	+
		-	++	-	-	-	++	+
	30	-	+	++	++	-	+	++
		-	+	-	+	-	+	++
	31	-	++	++	++	++	+	+
		-	+	-	-	-	+	-
	32	-	++	++	++	++	+	-
		-	-	-	+	-	-	-
	33	-	++	++	-	+	+++	+
		-	+	-	+	+	+++	+
	34	-	+	+	+	-	+	+
		-	+	-	-	-	+	+
	35	-	+	+	+++	+	+	+
		-	+	-	-	-	+	+
	36	-	+	+	++	+	+	+
		-	+	-	-	+	+	+
	37	-	++	++	++	++	+	+
		-	+	-	-	-	+	+
	38	-	++	+++	-	-	-	-
		-	-	-	-	-	-	-
	39	-	++	+	++	++	+	+
		-	+	+	-	-	+	+
	40	-	++	+++	++	++	+	+
		-	+	+	-	-	+	-
	41	-	++	-	+	+	++	++
		-	++	-	-	-	-	-
	42	-	++	-	+	+	+	++
		-	+	-	-	-	-	-
	43	-	++	++	++	-	++	+++
		-	-	-	-	-	-	-
	44	-	++	++	-	+	++	++
		-	++	-	+	+	-	-
	45	*-*	*++*	*+++*	*-*	*+*	*+*	*+*
		-	+	-	-	*+*	-	+
	46	-	+++	+++	-	-	++	++
		-	+	-	-	-	-	-
	47	-	++	-	++	-	+	++
		-	+	-	-	-	-	-
	48	-	++	++	-	+	+	+
		-	+	-	+	-	+	+
	49	-	+	++	++	++	+	+
		-	+	-	-	-	+	+
	50	-	+	+	++	+	+	+
		-	+	-	-	+	+	+
	51	-	+	+	+++	+	+	++
		-	+	-	-	-	+	++

+ low reactivity; ++ medium reactivity; +++ high reactivity; - non-reactivity.

**Table 4 toxins-12-00649-t004:** Relative intensity on immunoblots assay with bromelain and sera IgE profile.

Bromelain (CCD)					
Patient	Relative Intensity on Immunoblotting Reaction	sIgE (RAST)	Patient	Relative Intensity on Immunoblotting Reaction	sIgE (RAST)
1	+++	i1, i3, i70	27	-	i1, i3, i70
2	+	i1, i3, i70	28	-	i1, i3, i70
3	+	i1, i3, i70	29	-	i1, i3, i70
4	+	i1, i3, i70	30	-	i1, i3, i70
5	+++	i1, i3, i70	31	-	i1, i3
6	++	i1, i3, i70	32	-	i1, i3
7	+	i1, i3, i70	33	-	i1, i70
8	+	i1, i3, i70	34	-	i1, i70
9	+	i1, i3, i70	35	-	i1, i3
10	+	i1, i3, i70	36	-	i1, i70
11	++	i1, i3	37	-	i1
12	+++	i1, i70	38	-	i1
13	+	i1, i3	39	-	i1
14	+	i1, i3	40	-	i1
15	+	i3	41	-	i1
16	+++	i1	42	-	i70
17	+	i70	43	-	i70
18	+	i70	44	-	i70
19	+	i3	45	-	i70
20	+	i70	46	-	i70
21	++	i3	47	-	i3
22	++	i1	48	-	i3
23	-	i1, i3, i70	49	-	i3
24	-	i1, i3, i70	50	-	i1
25	-	i1, i3, i70	51	-	i70
26	-	i1, i3, i70			

i1: RAST for bee venom; i3: RAST for wasp venom; i70: RAST for ant venom, + weak reactivity; ++ intermediate reactivity; +++ high reactivity; - without reactivity.

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
