# Peer review of "Cross-Reactive Carbohydrate Determinant in Apis mellifera, Solenopsis invicta and Polybia paulista Venoms: Identification of Allergic Sensitization and Cross-Reactivity"

_toxins, 2020, doi:10.3390/toxins12100649_

Round 1

Reviewer 1 Report

Content of the article in propriate order, results well presented, collected measurement data sufficient to point analytical conclusions.

Fruitful discussion of results and clear conclusion.

Paper to accepted in presented form.

Author Response

Attending to the referee, we wish to express our thanks to the criticism and suggestions given to our manuscripts.

  1. The English language: text was reviewed
  2. All misspelt words and standardization of acronyms and terms, were reviewed

Reviewer 2 Report

The manuscript "Cross-Reactive Carbohydrate Determinant in Apis mellifera, Solenopsis invicta and Polybia paulista venoms: Identification of allergic sensitization and cross-reactivity" concerns an interesting issue on bromelain as a marker and CCD inhibitor to test sIgE reactivity. This could help to complement diagnostic methods and help to differentiate specific reactivity to allergen´s peptide epitopes from cross-reactivity caused by CCD, which is extremely important in clinical practice. The manuscript is technically sound, and conclusions are clear.

Author Response

Attending to the referee, we wish to express our thanks to the criticism and suggestions given to our manuscripts.

  1. The English language: text was reviewed

2. All misspelt words and standardization of acronyms and terms, were reviewed.

  1. Figure 1 legend was modified as recommended
  2. We maintained tables 2 and 3 in the body of the text and relocated figures 1A and 2A to the appendix
  3. We described with more detailed of recombinant allergens obtention and purification process in Material and Methods section 5.3, as recommended

Reviewer 3 Report

This manuscript is focused on identifying cross-reactivity to major venom proteins of Apis mellifera, Polybia paulista and Solenopsis invicta. There are several points to be corrected for full recommendation.

  1. Extensive editing of English language and style is essentially required.
  2. Change all the uncorrected words. (ex: Himinoptera to Hymenoptera in Line 80, 2.1.)
  3. Unify to use small letters or capital letters in naming the proteins in Figure1. (ex: Change the first letter into capital letter; 2: antigen 5 (Solenopsis invicta), 4: antigen 5 (Polybia paulista) and 6: recombinant PLA1)
  4. I recommend authors to move immunoblotting assay results (Figure 2 and 3) into supplementary section and summarize all the data into table. This might help comprehensively understand the whole data set.
  5. In Materials and Methods section (5.3., Line 309), there should be more descriptions about purifications. In venom immunotherapy related tests, especially checking cross-reactivity with antigens, one of the most important thing is to use highly purified recombinant venom protein. Therefore, I suggest authors to put additional description of purification processes of rPoly p1 and p5 in 5.3. section.

There are several points to be modified and corrected as below. (References section)

  1. Modify the scientific name to Italic style. (Polybia paulista in Line 418 and 422)
  2. Unify the letter style of paper title into small letters. (Line 453)

Author Response

Attending to the referee, we wish to express our thanks to the criticism and suggestions given to our manuscripts.

  1. The English language: text was reviewed
  2. All misspelt words and standardization of acronyms and terms were reviewed
  3. Figure 1 legend was modified as recommended
  4. We maintained tables 2 and 3 in the body of the text and relocated figures 1A and 2A to the appendix
  5. We described with more detailed of recombinant allergens obtention and purification process in Material and Methods section 5.3, as recommended
  6. As suggested, we revised all references